# Ebselen Inhibits the Growth of Lung Cancer Cells via Cell Cycle Arrest and Cell Death Accompanied by Glutathione Depletion

**DOI:** 10.3390/molecules28186472

**Published:** 2023-09-06

**Authors:** Woo Hyun Park

**Affiliations:** Department of Physiology, Medical School, Jeonbuk National University, 20 Geonji-ro, Deokjin, Jeonju 54907, Republic of Korea; parkwh71@jbnu.ac.kr

**Keywords:** ebselen, lung cancer cells, human pulmonary fibroblast, cell death, reactive oxygen species, glutathione

## Abstract

Ebselen is a glutathione (GSH) peroxidase (GPx) mimic originally developed to reduce reactive oxygen species (ROS). However, little is known about its cytotoxicological effects on lung cells. Therefore, this study aimed to investigate the effects of Ebselen on the cell growth and cell death of A549 lung cancer cells, Calu-6 lung cancer cells, and primary normal human pulmonary fibroblast (HPF) cells in relation to redox status. The results showed that Ebselen inhibited the growth of A549, Calu-6, and HPF cells with IC_50_ values of approximately 12.5 μM, 10 μM, and 20 μM, respectively, at 24 h. After exposure to 15 μM Ebselen, the proportions of annexin V-positive cells were approximately 25%, 65%, and 10% in A549, Calu-6, and HPF cells, respectively. In addition, Ebselen induced arrest at the S phase of the cell cycle in A549 cells and induced G2/M phase arrest in Calu-6 cells. Treatment with Ebselen induced mitochondrial membrane potential (MMP; ΔΨm) loss in A549 and Calu-6 cells. Z-VAD, a pan-caspase inhibitor, did not decrease the number of annexin V-positive cells in Ebselen-treated A549 and Calu-6 cells. Intracellular ROS levels were not significantly changed in the Ebselen-treated cancer cells at 24 h, but GSH depletion was efficiently induced in these cells. Z-VAD did not affect ROS levels or GSH depletion in Ebselen-treated A549 or Ebselen-treated Calu-6 cells. In conclusion, Ebselen inhibited the growth of lung cancer and normal fibroblast cells and induced cell cycle arrest and cell death in lung cancer cells with GSH depletion.

## 1. Introduction

The human lung is vulnerable to a wide array of injuries through airborne and bloodborne routes which are risk factors for lung fibrosis and cancer [1]. In a healthy lung, the natural repair process aims to restore normal structure and function. However, dysfunctional repair can lead to abnormal lung structure and function. During pathologic lung repair, pulmonary fibroblast (PF) cells are insufficient or accumulated unnecessarily, resulting in abnormalities in tissue function and eventually lung diseases [2]. Lung cancer, a widespread disease, ranks among the top causes of cancer-related deaths globally [3,4]. It is broadly categorized as small cell lung cancer (SCLC) and non-small cell lung cancer (NSCLC), accounting for approximately 15% and 85% of cases, respectively [3,4]. Ongoing studies on cytotoxic drugs have illuminated avenues for improved lung cancer treatment strategies. Among the chemotherapeutic approaches being tried are those targeting apoptosis and necrosis. Apoptosis and necrosis represent cellular responses triggered by cytotoxic agents [5,6]. The inauguration of apoptosis in the mitochondrial pathway is prompted or accompanied by the loss of mitochondrial membrane potential (MMP; ΔΨm) [5]. The eukaryotic cell cycle consists of four distinct phases: the G1 phase, the S phase, the G2 phase, and the M phase [7,8]. Regulation of the cell cycle involves procedures crucial to cell survival and cell proliferation, including the recognition and repair of genetic impairment as well as the restriction of uncontrolled cell division [7,8]. Therefore, the focused suppression of anti-apoptotic pathways and effective regulation of cell cycle advancement remain appealing objectives in devising successful approaches for lung cancer therapy.

Reactive oxygen species (ROS) are highly unstable molecules generated during mitochondrial oxidative phosphorylation and produced by enzymes such as nicotinamide adenine dinucleotide phosphate (NADPH) oxidase and xanthine oxidase [9]. They include hydrogen peroxide (H_2_O_2_), hydroxyl radicals (^•^OH), and superoxide anions (O_2_^•−^). To redress the harmful effects of ROS, cells utilize several mechanisms that provide a balance between production and removal of ROS. Among them is the metabolism of H_2_O_2_ to O_2_ and H_2_O by catalase or glutathione (GSH) peroxidase (GPx) [10]. GSH is a vital nonprotein antioxidant that is known to protect cells from toxic insults [11]. In addition, thioredoxin (Trx, a redox regulatory protein), Trx reductase (TrxR), and NADPH systematically regulate redox reactions in biosynthetic pathways and control redox homeostasis [12,13]. Modulation of the Trx system is thus a promising target for cancer therapy [12,13]. The imbalance between ROS generation and their neutralization due to excess ROS production or reduced antioxidant response leads to oxidative stress, which affects the progression of many diseases in a variety of ways [14,15].

Ebselen (2-Phenyl-1,2-benzisoselenazol-3(2H)-one) is a seleno-organic compound that was originally developed as a GPx mimic to remove ROS [13,16,17,18]. The GPx activity of Ebselen is well characterized, in that the Se-N bond in Ebselen is easily cleaved by thiols to the corresponding selenol to reduce hydrogen/lipid peroxides [13,16,17,18]. However, selenium from Ebselen is not incorporated into GPx [19,20]. In addition, Ebselen is an excellent substrate for mammalian TrxR in various species [21]. Numerous studies suggest that the biological effects of Ebselen are highly related to the Trx system [13,22]. Owing to its antioxidant properties, Ebselen can alleviate oxidative stress in cells and tissues, restore cells from ROS-induced damage, and potentially prevent diseases caused by oxidative stress. Ebselen has received attention due to its prospect as a therapeutic drug for various diseases [13,16,23,24,25]. Furthermore, Ebselen has demonstrated its protective efficacy against bleomycin-induced lung fibrosis [26] and has restored vascular function during influenza A virus-induced exacerbations of cigarette smoke-induced lung inflammation [27].

Despite this background, there is a lack of research addressing the precise molecular mechanisms responsible for Ebselen’s cytotoxic and anti-growth effects on cells, particularly in the context of lung cancer. The impact of Ebselen on cell growth, cell death, and the cellular redox state in lung cells needs further clarification. Therefore, the primary objective of this study was to investigate the underlying molecular mechanisms behind Ebselen’s anti-growth effects, focusing on cell death, cell cycle arrest, as well as changes in ROS and GSH levels. This investigation utilized A549 and Calu-6 lung cancer cells. Additionally, the study aimed to examine the influence of Ebselen on cell growth and cell death in primary normal human PF (HPF) cells.

## 2. Results

### 2.1. Effects of Ebselen on Lung Cancer and Normal Cell Growth

The effects of Ebselen on the growth of lung cells were observed using MTT assays. The growth of A549 lung cancer cells was inhibited by Ebselen with a half-maximal inhibitory concentration (IC_50_) of ~12.5 μM after 24 h incubation (Figure 1A). Treatment with 15 μM Ebselen decreased the growth of A549 cells by approximately 65% (Figure 1A). A dose-dependent reduction in cell growth was observed in Calu-6 lung cancer cells with an IC_50_ of ~10 μM following treatment with Ebselen for 24 h (Figure 1B). At a concentration of 15 μM, Ebselen appeared to reduce the growth of Calu-6 cells by approximately 90% (Figure 1B). Ebselen inhibited the growth of primary normal HPF cells with an IC_50_ of ~20 μM after 24 h incubation (Figure 1C). Treatment with 15 μM Ebselen reduced the growth of primary HPF cells by approximately 10% (Figure 1C).

### 2.2. Effects of Ebselen on Apoptosis of Lung Cancer and HPF Cells

The effects of Ebselen on cell death via apoptosis were evaluated using annexin V-FITC and/or PI-stained cells. Viable cells are negative for both annexin V-FITC and PI; apoptotic cells are positive for annexin V-FITC and negative for PI, whereas late apoptotic dead cells display both high annexin V-FITC and PI labeling. Non-viable cells that have undergone necrotic cell death are positive for PI and negative for annexin V-FITC. As shown in Figure 2A,B, Ebselen at 2–5 μM did not affect the number of annexin V-positive A549 cells at 24 h, whereas 10–20 μM Ebselen significantly increased the number of annexin V-positive cells. After exposure to 15 μM and 20 μM Ebselen, proportions of annexin V-positive cells were approximately 25% and 75% in A549 cells, respectively (Figure 2A,B). At all tested concentrations, Ebselen increased the number of annexin V-positive Calu-6 cells. Ebselen at 10 μM and 15 μM increased the number of annexin V-positive cells by approximately 45% and 65%, respectively (Figure 2A,C). Calu-6 cells treated with Ebselen showed relatively high proportions of PI-positive and annexin V-FITC-negative cells (Figure 2A,C). In addition, 2–10 μM Ebselen did not increase the number of annexin V-positive HPF cells at 24 h, whereas 15 M and 20 μM Ebselen significantly increased the number of annexin V-positive cells by approximately 10% and 45%, respectively (Figure 2A,D). These results underscore the concentration-dependent apoptotic effects of Ebselen on lung cancer cells, emphasizing its influence on cell growth.

### 2.3. Effects of Ebselen on Sub-G1 and Cell Cycle Distributions in Lung Cancer Cells

The observed growth inhibition effect of Ebselen on A549 and Calu-6 lung cancer cells can be attributed to its impact on cell cycle progression. To gain insights from a cell growth perspective, the distribution of cells across various phases of the cell cycle was analyzed. DNA flow cytometric analysis indicated that 2–10 μM Ebselen did not increase the number of sub-G1 phase A549 cells at 24 h but Ebselen at 15 μM and 20 μM significantly increased the number of these cells at 24 h by approximately 25% and 45%, respectively (Figure 3A,B). Furthermore, 2–10 μM Ebselen did not increase the number of sub-G1 phase Calu-6 cells, whereas 15 μM Ebselen significantly increased the number of these cells by approximately 9% at 24 h (Figure 3A,C). As shown in Figure 3A,D, 20 μM Ebselen significantly induced cell cycle arrest of A549 cells in the G1 phase, while Ebselen at all tested concentrations induced S phase arrest, and 15 μM and 25 μM Ebselen significantly increased the number of S phase cells. In contrast, 15 μM and 25 μM Ebselen significantly decreased the number of A549 cells in the G2/M phase (Figure 3A,D). In Ebselen-treated Calu-6 cells, the proportions of G1 phase cells generally decreased, whereas the proportions of G2/M phase cells increased (Figure 3A,E). At concentrations of 10 μM and 15 μM, Ebselen significantly induced a G2/M phase arrest of Calu-6 cells (Figure 3A,E). Taken together, these results demonstrate that Ebselen exerts its growth-inhibitory effects on lung cancer cells through a profound impact on cell cycle regulation, ultimately influencing their growth or proliferation.

### 2.4. Effects of Ebselen on Mitochondrial Membrane Potential (MMP; ΔΨm) in Lung Cancer Cells

Apoptosis and necrosis are closely related to MMP (ΔΨm) loss. Therefore, the loss of MMP (ΔΨm) in Ebselen-treated cells was evaluated using rhodamine 123 dye. While 2–10 μM Ebselen did not affect MMP (ΔΨm) loss in A549 cells, Ebselen at 15 μM and 20 μM significantly increased MMP (ΔΨm) loss cells by approximately 20% and 80%, respectively (Figure 4A). In addition, 2–5 μM Ebselen did not increase MMP (ΔΨm) loss in Calu-6 cells, whereas 10 μM and 15 μM Ebselen increased MMP (ΔΨm) loss cell by approximately 55% and 70%, respectively (Figure 4B).

### 2.5. Effects of Z-VAD on Cell Growth and Cell Death in Ebselen-Treated Lung Cancer Cells

The effects of Z-VAD-FMK, a pan-caspase inhibitor, on cell growth and cell death in Ebselen-treated A549 and Calu-6 cells were examined at 24 h. Based on previous experiments related to caspase inhibitors [28], A549 and Calu-6 cells were pretreated with Z-VAD at a concentration of 15 μM for 1 h before 15 μM or 10 μM Ebselen treatment. This concentration of Ebselen was appropriate for distinguishing alterations in cell growth and death. Z-VAD significantly attenuated the growth inhibition of Ebselen-treated A549 cells (Figure 5A). However, Z-VAD did not significantly affect the growth of Ebselen-treated Calu-6 cells (Figure 5B). Z-VAD slightly increased the number of annexin V-positive cells in Ebselen-treated A549 cells but did not affect the number of annexin V-positive cells in Ebselen-treated Calu-6 cells (Figure 5C,D).

### 2.6. Effects of Z-VAD on ROS and GSH Levels in Ebselen-Treated Lung Cancer Cells

The effects of Z-VAD on ROS and GSH levels in Ebselen-treated A549 and Calu-6 cells were investigated at 24 h. To assess the intracellular levels of ROS and GSH in Ebselen-treated cells, H_2_DCFDA and CMF fluorescent dyes were used for non-specific ROS and GSH, respectively. As shown in Figure 6A, there was no significant change in the intracellular DCF (ROS) levels in A549 cells treated with 15 μM Ebselen at 24 h. Z-VAD seemed to increase the DCF (ROS) level in Ebselen-treated and -untreated A549 cells (Figure 6A). It was observed that 10 μM Ebselen slightly increased DCF (ROS) levels in Calu-6 cells at 24 h, but that the effect of Z-VAD on DCF (ROS) levels in Ebselen-treated Calu-6 cells was insignificant (Figure 6B). Concerning GSH levels, 15 μM and 10 μM Ebselen increased the number of GSH-depleted cells in A549 and Calu-6 cells by approximately 19% and 38%, respectively (Figure 6C,D). Z-VAD appeared to increase the number of GSH-depleted Ebselen-treated A549 cells but did not alter the number of GSH-depleted Ebselen-treated Calu-6 cells (Figure 6C,D).

## 3. Discussion

As a promising novel drug with low toxicity and good bioavailability, Ebselen has elicited interest and is currently being evaluated as a potential therapeutic agent for several diseases, including those associated with lung health [13,16,22,23,24,25,26,27]. However, little is known about the cytotoxicological mechanisms of Ebselen’s action in the lung cells. Thus, the present study focused on elucidating the cytotoxicological effects of Ebselen on cell growth, cell death, and cell cycle distribution in lung cancer cells and normal HPF cells in relation to ROS and GSH.

Treatment with Ebselen decreased the growth of A549 and Calu-6 lung cancer cells with IC_50_ values of approximately 12.5 µM and 10 μM, respectively, at 24 h. In addition, Ebselen inhibited the growth of primary normal HPF cells with an IC_50_ of ~20 μM at 24 h. Treatment with 15 μM Ebselen decreased the growth of A549, Calu-6, and HPF cells by approximately 65%, 90%, and 10%, respectively. After exposure to 15 μM Ebselen, proportions of annexin V-positive cells were approximately 25%, 65%, and 10% in A549, Calu-6, and HPF cells, respectively. Therefore, among the cells tested, Calu-6 cells showed the highest susceptibility to Ebselen, while normal HPF cells were relatively resistant. In addition, cell growth inhibition was higher in Ebselen-treated lung cancer cells than in Ebselen-treated normal HPF cells. The differences between these cells could be attributed to different antioxidant enzyme capacities and basal mitochondrial activities. Treatment with 20 μM Ebselen induced a significant level of cycle cell arrest among A549 cells in the G1 phase and all the Ebselen concentrations tested induced S phase arrest. In Ebselen-treated Calu-6 cells, the number of cells increased in the G2/M phase. Therefore, Ebselen exerts its inhibitory effects on cell growth by inducing G1- and/or S phase arrests in A549 cells and G2/M phase arrest in Calu-6 cells. Z-VAD significantly attenuated growth inhibition in Ebselen-treated A549 cells but not in Ebselen-treated Calu-6 cells, implying that caspases are involved in cell growth and/or cell cycle progression in A549 cells.

There were increased percentages of sub-G1 cells in A549 and Calu-6 cells with higher concentrations of Ebselen. However, the number of annexin V-positive cells was higher than the number of sub-G1 cells in both A549 and Calu-6 Ebselen-treated cells; 10 μM Ebselen in A549 cells did not increase the number of sub-G1 cells but significantly increased the number of annexin V-positive cells. In Calu-6 cells, 2–10 μM Ebselen did not increase the number of sub-G1 cells but significantly increased the number of annexin V-positive cells. Similarly, Ebselen at 10 μM and 15 μM in Calu-6 cells increased the number of annexin V-positive cells by 45% and 65%, respectively, but increased the number of sub-G1 cells only by approximately 1% and 9%, respectively. Therefore, Ebselen is likely to induce lung cancer cell death via apoptosis. Additionally, Calu-6 cells treated with Ebselen showed high proportions of PI-positive and annexin V-negative cells. These results imply that Ebselen induced necrosis as well as apoptosis in Calu-6 cells. Since apoptosis and necrosis are closely related to the failure of MMP (ΔΨm) [29,30,31], relatively high doses of Ebselen strongly increased the loss of MMP (ΔΨm) in both lung cancer cells. As expected, the degree of MMP (ΔΨm) loss cells in 10–15 μM Ebselen-treated A549 cells was lower than that in Calu-6 cells treated with the same dose. The degree of MMP (ΔΨm) loss cells in Ebselen-treated lung cells was similar to that of annexin V-positive cells. Interestingly, Z-VAD slightly increased annexin V-positive cells in Ebselen-treated A549 cells, but did not change annexin V-positive cells in Ebselen-treated Calu-6 cells. We previously reported that Z-VAD significantly decreased the number of annexin V-positive cells in A549 or Calu-6 cells treated with numerous agents [31,32,33,34,35]. Therefore, Ebselen seems to induce caspase-independent apoptosis in lung cancer cells, especially A549 cells.

Previous studies have reported that Ebselen reduces hydroperoxides such as H_2_O_2_, phospholipid hydroperoxide, and cholesterol ester hydroperoxide [13,16,17,18]. As an excellent substrate of mammalian TrxR, it affects the Trx system [13,21,22]. High concentrations of Ebselen may disturb the Trx system by competing with electrons from Trx and *NADPH*, consequently leading to cell dysfunction and death [21,36]. Results from the present study did not show a significant change in intracellular DCF (ROS) levels in 15 μM Ebselen-treated A549 cells at 24 h and showed a slight increase in 10 μM Ebselen-treated Calu-6 cells. Z-VAD did not affect DCF (ROS) levels significantly in Ebselen-treated A549 or Calu-6 cells. These results suggest that changes in DCF (ROS) levels induced by Ebselen are not correlated with the death of lung cancer cells. Previous studies have reported that intracellular GSH content is inversely related to the induction of cell death [37,38,39]. A decrease in GSH content is part of the sequence of events inducing apoptosis, and its depletion occurs when the MMP (ΔΨm) is completely disrupted [40]. In this study, there was an increased number of GSH-depleted cells in Ebselen-treated A549 and Calu-6 cells. Furthermore, Z-VAD, which showed no anti-apoptotic effect in both cancer cells treated with Ebselen, did not decrease the number of GSH-depleted cells in these cells. These current results are in agreement with previous studies which reported that the induction of cell death is strongly correlated to the depletion of GSH content [37,38,39,41]. Numerous studies suggest that the cellular effect of Ebselen could be predominant via the Trx system rather than via the GSH-related system [13,22]; it is therefore worth studying the cytotoxicological effects of Ebselen through Trx and TrxR alterations further.

In summary, it is hypothesized that Ebselen treatment not only suppressed the proliferation of A549 and Calu-6 lung cancer cells and HPF normal cells but also triggered cell cycle arrest and cell death through apoptosis and/or necrosis in lung cancer cells. This effect appeared to be linked to GSH depletion rather than an increase in ROS levels (Figure 7). Further investigations are necessary to delve into the relationship between GSH depletion and loss of MMP (ΔΨm), as well as the mechanisms by which alterations in GSH levels influence the cell cycle. Moreover, obtaining a more comprehensive understanding of Ebselen’s molecular cytotoxicity could entail examining the differences in cell signaling between lung cancer cells and normal lung cells. In conclusion, the findings from this study shed light on the cytotoxic effects of Ebselen on lung cancer and normal cells in terms of inhibiting cell growth, inducing cell death, and affecting cellular redox status.

## 4. Materials and Methods

### 4.1. Cell Culture

Human NSCLC A549 cells and SCLC Calu-6 cells were obtained from the American Type Culture Collection (Manassas, VA, USA). Primary normal HPF cells were purchased from PromoCell GmbH (C-12360, Heidelberg, Germany) and used between passages five and seven. These lung cells were kept in a standard humidified incubator at 37 °C with 5% CO_2_ and cultured in RPMI-1640 medium supplemented with 10% fetal bovine serum (Sigma-Aldrich Co., St. Louis, MO, USA) and 1% penicillin-streptomycin (Gibco BRL, Grand Island, NY, USA). Cells were grown in 100 mm plastic cell culture dishes (BD Falcon, Franklin Lakes, NJ, USA) and harvested with trypsin-EDTA (Gibco BRL).

### 4.2. Reagents

Ebselen (2-phenyl-1,2-benzisoselenazol-3(2H)-one), obtained from Sigma-Aldrich Co. (CAS: 60940-34-3), is a compound with the chemical formula C13H9NOSe and a molecular weight of 274.18. Its purity level is indicated as greater than or equal to 98% according to TLC analysis. Ebselen was dissolved in dimethyl sulfoxide (DMSO; Sigma-Aldrich Co.) for experimental use at 100 mM as a stock solution. Z-VAD-FMK (benzyloxycarbonyl-Val-Ala-Asp-fluoromethylketone) was purchased from R&D Systems, Inc. (Minneapolis, MN, USA) and dissolved in DMSO to generate a 10 mM stock solution. Cells were pretreated with Z-VAD for 1 h prior to treatment with Ebselen. DMSO (0.1%) was used as a vehicle control. All stock solutions were wrapped in foil and kept at 4 °C or −20 °C.

### 4.3. Cell Growth Inhibition Assay

The effects of Ebselen on the growth of lung cancer and normal cells were determined using 3-(4,5-dimethylthiazol-2-yl)-2,5-diphenyltetrazolium bromide (MTT, Sigma-Aldrich Co.) assays, as previously described [42]. Briefly, cells were seeded into 96-well microtiter plates (Nunc, Roskilde, Denmark) at a density of 5 × 10^4^ cells/well. After incubation with designated doses of Ebselen (2–20 μM) and/or Z-VAD (15 μM) for 24 h, 20 μL of MTT solution (2 mg/mL in phosphate-buffered saline (PBS; GIBCO BRL)) was added to each well. Plates were then incubated at 37 °C for 4 h. Medium in each well was removed by pipetting. Then, 100–200 μL of DMSO was added to each well to solubilize the formazan crystals. Optical density was measured at 570 nm with a microplate reader (Synergy™ 2, BioTekR Instruments Inc. Winooski, VT, USA).

### 4.4. Annexin V-Fluorescein Isothiocyanate Staining (FITC) and Propidium Iodide (PI) Staining for Apoptotic Cell Detection

Apoptosis was identified via annexin V-FITC (Life Technologies, Carlsbad, CA, USA; Ex/Em = 488/519 nm) and PI staining (Sigma-Aldrich Co.; Ex/Em = 488 nm/617 nm), as previously described [43]. Briefly, 1 × 10^6^ cells in 60 mm culture dishes (BD Falcon) were preincubated with Z-VAD (15 μM) for 1 h prior to treatment with Ebselen at the indicated concentrations (2–20 μM) for 24 h. Cells were washed twice with cold PBS and then suspended in 200 μL of binding buffer (10 mM HEPES/NaOH pH 7.4, 140 mM NaCl, 2.5 mM CaCl_2_) at a density of 5 × 10^5^ cells/mL at 37 °C for 30 min. After adding annexin V-FITC (2 μL) and PI (1 μg/mL), cells were analyzed with a FAC Star flow cytometer (BD Sciences, Franklin Lakes, NJ, USA).

### 4.5. Cell Cycle and Sub-G1 Cell Analysis

Cell cycle and sub-G1 distributions of cells were analyzed using PI staining, as previously described [43]. Briefly, 1 × 10^6^ cells in 60 mm culture dishes (BD Falcon) were incubated with Ebselen at the indicated concentration (2–20 μM) for 24 h. Cells were washed with PBS and then incubated with 10 μg/mL PI and RNase (Sigma-Aldrich Co.) at 37 °C for 30 min. The proportions of cells in different phases of the cell cycle or with sub-G1 DNA content were measured and analyzed with a FAC Star flow cytometer (BD Sciences).

### 4.6. Measurement of MMP (ΔΨm)

MMP (ΔΨm) was examined using a cell-permeable and cationic rhodamine 123 fluorescent dye (Sigma-Aldrich Co.; Ex/Em = 485/535 nm). Briefly, 1 × 10^6^ cells in 60 mm culture dishes (BD Falcon) were incubated with Ebselen at the indicated concentrations (2–20 μM) for 24 h. Cells were washed twice with PBS and incubated with rhodamine 123 (0.1 mg/mL) at a concentration of 5 × 10^5^ cells/mL at 37 °C for 30 min. Rhodamine 123 staining intensities were determined using a FAC Star flow cytometer (BD Sciences). Rhodamine 123-negative (−) cells designated MMP (ΔΨm) loss cells.

### 4.7. Determination of Intracellular ROS Levels

Intracellular ROS, such as H_2_O_2_, ^•^OH, and ONOO^•^, were measured using the oxidation-sensitive fluorescent probe dye 2′,7′-dichlorodihydrofluorescein diacetate (H_2_DCFDA, Ex/Em = 495 nm/529 nm; Invitrogen Molecular Probes, Eugene, OR, USA), as previously described [42]. In brief, 1 × 10^6^ cells in 60 mm culture dishes (BD Falcon) were pretreated with Z-VAD (15 μM) for 1 h and then treated with the indicated amount of Ebselen for 24 h. Then, the cells were washed in PBS and incubated with 20 µM H_2_DCFDA at 37 °C for 30 min. The mean DCF fluorescence value was detected using a FAC Star flow cytometer (BD Sciences). The mean DCF levels were expressed as percentages compared to the control cells.

### 4.8. Detection of Intracellular GSH Levels

Cellular GSH levels were evaluated using 5-chloromethylfluorescein diacetate (CMFDA, Ex/Em = 522 nm/595 nm; Invitrogen Molecular Probes), as previously described [42]. In brief, 1 × 10^6^ cells in 60 mm culture dishes (BD Falcon) were pretreated with Z-VAD (15 μM) for 1 h and then treated with the indicated amount of Ebselen for 24 h. Then, the cells were washed with PBS and incubated with 5 µM CMFDA at 37 °C for 30 min. CMF fluorescence intensity was determined using a FAC Star flow cytometer (BD Sciences). CMF negative (−) staining indicated the depletion of GSH content in cells.

### 4.9. Statistical Analysis

The results show the mean of two or three independent experiments (mean ± SD). The data were analyzed using Instat software (GraphPad Prism 5.0, San Diego, CA, USA). A Student’s *t*-test or one-way analysis of variance with post hoc analysis using Tukey’s multiple comparison test was applied to determine statistical significance, which was defined as *p*-values of < 0.05.

## Figures and Tables

**Figure 1 molecules-28-06472-f001:**
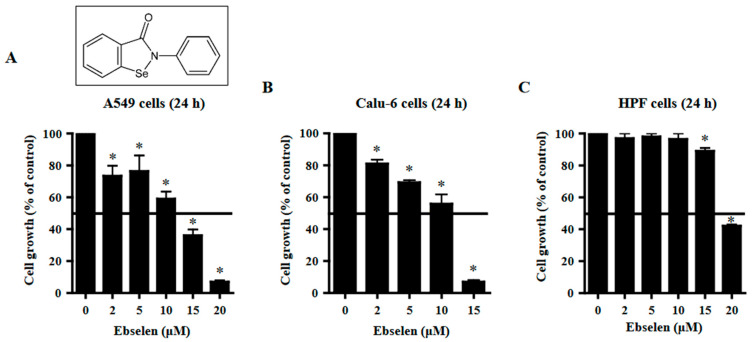
Effects of Ebselen on cell viability of lung cancer and normal cells. Exponentially growing cells were incubated with Ebselen at the indicated concentrations for 24 h. Cell growth was evaluated by MTT assays. The chemical figure indicates the structure of Ebselen (2-Phenyl-1,2-benzisoselenazol-3(2H)-one). Graphs show cell growth of Calu-6 cancer cells (**A**), A549 cancer cells (**B**), and primary normal HPF cells (**C**). * *p* < 0.05 compared to Ebselen-untreated control cells.

**Figure 2 molecules-28-06472-f002:**
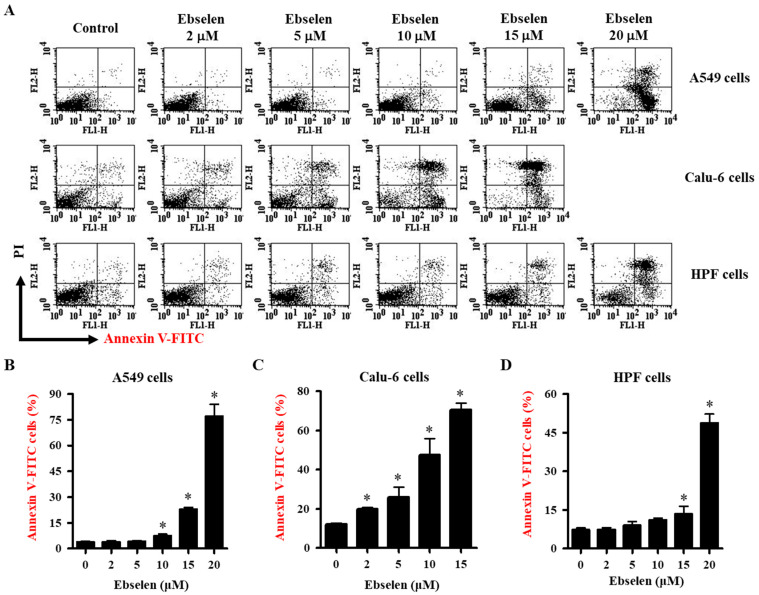
Effects of Ebselen on apoptosis in lung cancer and normal cells. Cells in exponential growth phase were incubated with Ebselen at the indicated concentrations for 24 h. Annexin V-FITC and PI staining cells were evaluated with a FAC Star flow cytometer. (**A**) Representative figures for annexin V-FITC and PI staining cells in lung cancer and normal cells. (**B**–**D**) Graphs show proportions of annexin V-positive cells derived from Figure (**A**) in Calu-6 (**B**), A549 (**C**), and HPF cells (**D**). * *p* < 0.05 compared to Ebselen-untreated control cells.

**Figure 3 molecules-28-06472-f003:**
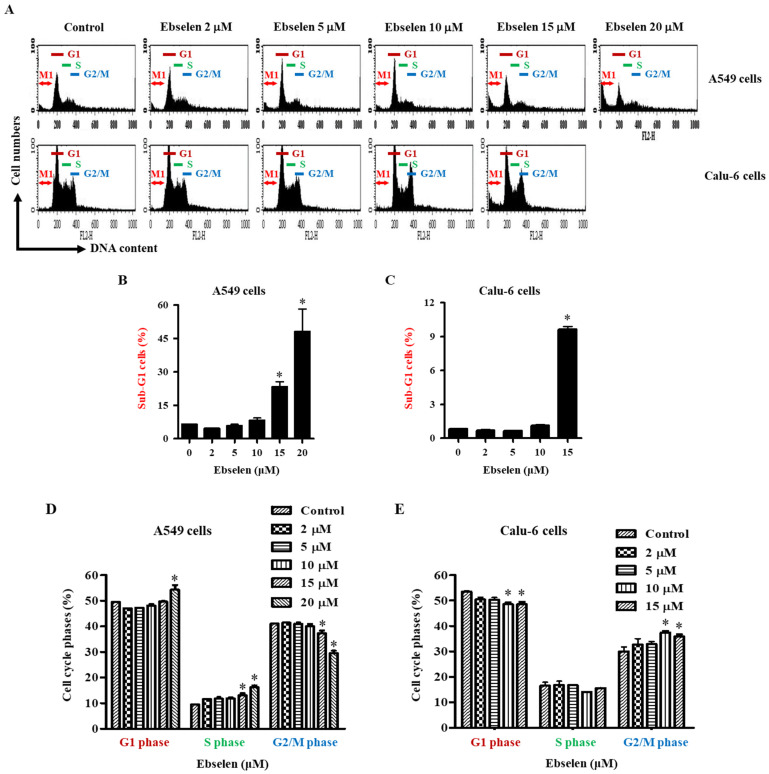
Effects of Ebselen on sub-G1 and cell cycle phase distributions in A549 and Calu-6 cancer cells. Cells in the exponential growth phase were incubated with Ebselen at the indicated concentrations for 24 h. Cell cycle phase distributions were evaluated by DNA flow cytometry. (**A**) Each histogram shows the cell cycle distributions in Ebselen-treated A549 and Calu-6 lung cancer cells. M1 indicates sub-G1 cells. G1, S, and G2 represent each phase of the cell cycle. (**B**,**C**): Graphs show the proportions of sub-G1 cells derived from M1 in Figure (**A**) in A549 (**B**) and Calu-6 cells (**C**). (**D**,**E**) Graphs show cell cycle phase distributions derived from G1, S, and G2 in Figure (**A**) in A549 (**D**) and Calu-6 cells (**E**). * *p* < 0.05 compared to Ebselen-untreated control cells.

**Figure 4 molecules-28-06472-f004:**
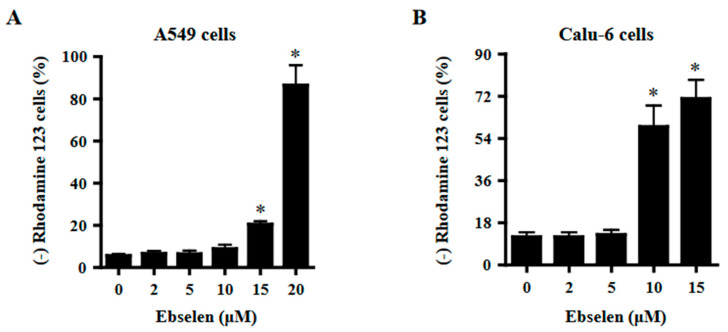
Effects of Ebselen on MMP (ΔΨm) levels in A549 and Calu-6 cancer cells. Exponentially growing cells were incubated with Ebselen at the indicated concentrations for 24 h. MMP (ΔΨm) in lung cancer cells was measured using a FAC Star flow cytometer. Graphs show proportions of rhodamine 123-negative [MMP (ΔΨm) loss] A549 (**A**) and Calu-6 cells (**B**). * *p* < 0.05 compared to Ebselen-untreated control cells.

**Figure 5 molecules-28-06472-f005:**
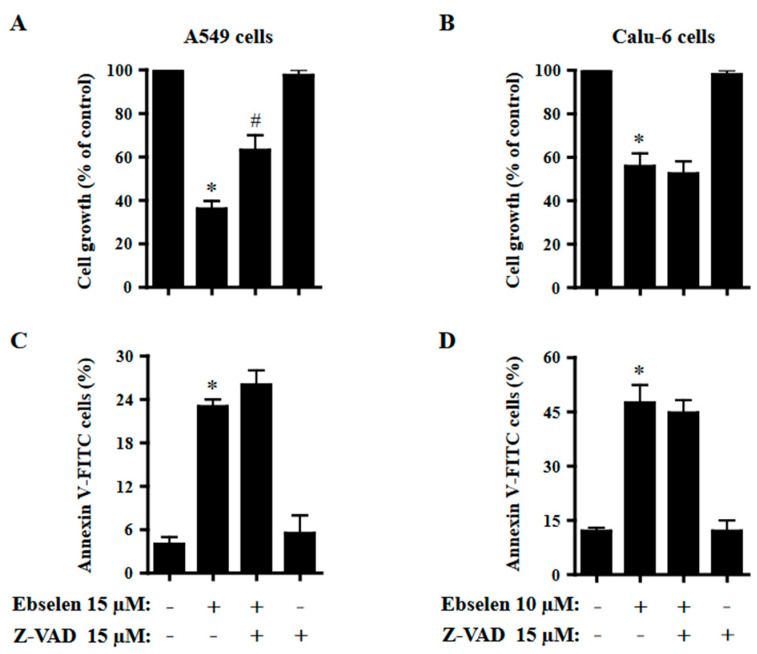
Effects of Z-VAD on cell viability and cell death in Ebselen-treated A549 and Calu-6 lung cancer cells. Exponentially growing cells were pretreated with Z-VAD (15 μM) for 1 h and then treated with 15 µM or 10 μM Ebselen for 24 h. (**A**,**B**) Graphs show cell growth of A549 (**A**) and Calu-6 cells (**B**), evaluated by MTT assays. (**C**,**D**) Graphs show proportions of annexin V-positive A549 (**C**) and Calu-6 cells (**D**), measured with a FAC Star flow cytometer. * *p* < 0.05 compared to Ebselen-untreated control cells. # *p* < 0.05 compared to cells treated with Ebselen only.

**Figure 6 molecules-28-06472-f006:**
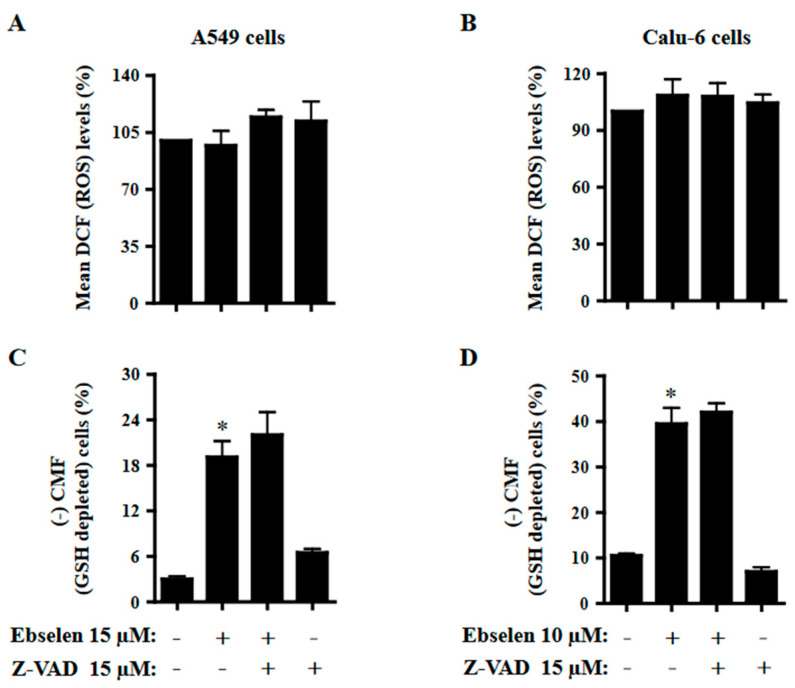
Effects of Z-VAD on ROS and GSH levels in Ebselen-treated A549 and Calu-6 lung cancer cells. Exponentially growing cells were pretreated with Z-VAD (15 μM) for 1 h and then treated with 15 µM or 10 μM Ebselen for 24 h. Intracellular DCF (ROS) and CMF (GSH) levels in lung cancer cells were measured using a FAC Star flow cytometer. (**A**,**B**) Graphs indicate the mean DCF (ROS) levels (%) in A549 (**A**) and Calu-6 cells (**B**). (**C**,**D**) Graphs indicate the percentages of (-) CMF (GSH-depleted) A549 (**C**) and Calu-6 cells (**D**). * *p* < 0.05 compared to Ebselen-untreated control cells.

**Figure 7 molecules-28-06472-f007:**
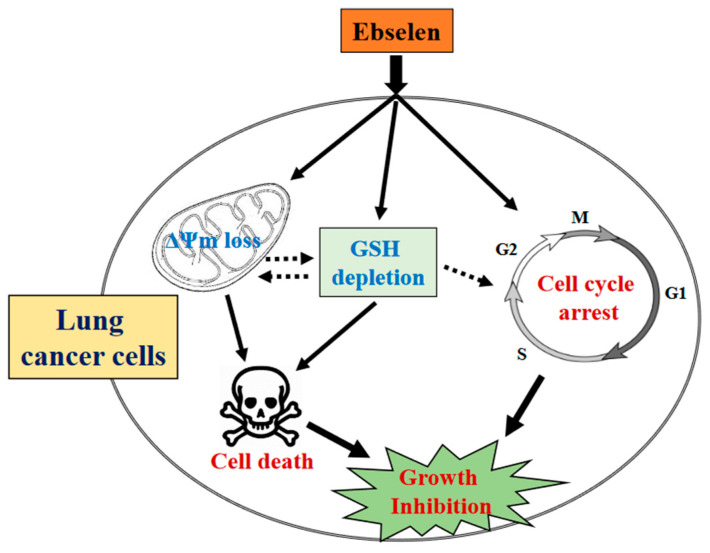
Schematic diagram of Ebselen-induced growth inhibition in lung cancer cells.

## Data Availability

Data collected during the present study are available from the corresponding author upon reasonable request.

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
