# Peer review of "Ebselen Inhibits the Growth of Lung Cancer Cells via Cell Cycle Arrest and Cell Death Accompanied by Glutathione Depletion"

_molecules, 2023, doi:10.3390/molecules28186472_

Round 1
Reviewer 1 Report
The novelty of paper is not enough. It has only in vitro results and week mechanism. Therefore, this paper is rejected.
Extensive editing of English language required
Author Response
The novelty of paper is not enough. It has only in vitro results and week mechanism. Therefore, this paper is rejected. Extensive editing of English language required
>>> Thank you for your valuable comments and evaluations. I tried to change the manuscript to comply with the criticism raised by another reviewer II. In addition, English language proofreading and editing were undertaken by the journal's English language experts, based on the suggestions and recommendations of the editors and reviewers. I did my best to ensure that the manuscript meets the standards required for acceptance, and I greatly appreciate your continued consideration and support.
I appreciate Editor and Reviewer for their considerate cooperation.
Reviewer 2 Report
The starting point of this paper is good, but there are still many questions that need to be addressed in order to better serve the readers:
1. Although the compounds described in this paper are commercially available products, characterization and purity analysis should also be provided.
2. It is suggested that the author use color images, as it would better serve the readers.
3. There is excessive common knowledge description in the introduction and it should be condensed.
4. The experiments mentioned in the text do not fully support the mechanism described in Figure 7; therefore, it is recommended to conduct further mechanism research and at least address issues such as lack of comparison groups for upstream and downstream pathways.
5. Such as commonly used antioxidants like ascorbic acid whether they also have an effect or not?
Author Response
The starting point of this paper is good, but there are still many questions that need to be addressed in order to better serve the readers:
>>> Thank you for your valuable comments and positive evaluations. I tried to change the manuscript to comply with the criticism raised by the reviewer II. In addition, English language proofreading and editing were undertaken by the journal's English language experts, based on the suggestions and recommendations of the editors and reviewers.
- Although the compounds described in this paper are commercially available products, characterization and purity analysis should also be provided.
>>> Thank you very much for your comment. I described the properties and purity analysis of Ebselen in Materials and Methods section of the new version of manuscript.
2. Materials and Methods
2.2. Reagents
“Ebselen [2-phenyl-1,2-benzisoselenazol-3(2H)-one], obtained from Sigma-Aldrich Co. (CAS: 60940-34-3), is a compound with the chemical formula C13H9NOSe and a molecular weight of 274.18. Its purity level is indicated as greater than or equal to 98% according to TLC analysis. Ebselen was dissolved in dimethyl sulfoxide (DMSO; Sigma-Aldrich Co.) for experimental use at 100 mM as a stock solution…..”
- It is suggested that the author use color images, as it would better serve the readers.
>>> Thank you for your valuable comments. Your thoughtful comments will greatly help our readers understand the results easily. Therefore, the colored lines and letters in Figure 2 and Figure 3 of the new version of manuscript were used to help track the results.
- There is excessive common knowledge description in the introduction and it should be condensed.
>>> Thank you for your thoughtful comments. Although this manuscript has been reviewed by an English language editing company, attempts have been made throughout the manuscript to comprehensively explain or paraphrase sentences. In particular, in the Introduction part, unnecessary or too general parts were deleted to provide concise and clean explanations. Your thoughtful comments will be of great help in writing future papers.
- The experiments mentioned in the text do not fully support the mechanism described in Figure 7; therefore, it is recommended to conduct further mechanism research and at least address issues such as lack of comparison groups for upstream and downstream pathways.
>>> Thank you for your insightful comments. I appreciate your observation that the mentioned experiment may not completely match the mechanism described in Figure 7. Your recommendation to conduct further mechanism research is duly noted. Addressing issues such as the absence of comparison groups for upstream and downstream pathways is indeed an important consideration for ensuring a comprehensive understanding of the mechanisms involved. I have described in detail the contents of these additional studies in the Discussion part of the new version of manuscript. I value your input and will certainly take your suggestions into account for our ongoing research and analysis.
- Discussion
………“In summary, it is hypothesized that Ebselen treatment not only suppressed the proliferation of A549 and Calu-6 lung cancer cells and HPF normal cells but also triggered cell cycle arrest and cell death through apoptosis and/or necrosis in lung cancer cells. This effect appeared to be linked to GSH depletion rather than an increase in ROS levels (Figure 7). Further investigations are necessary to delve into the relationship between GSH depletion and loss of MMP (∆Ψm), as well as the mechanisms by which alterations in GSH levels influence the cell cycle. Moreover, obtaining a more comprehensive understanding of Ebselen's molecular cytotoxicity could entail examining the differences in cell signaling between lung cancer cells and normal lung cells. In conclusion, the findings from this study shed light on the cytotoxic effects of Ebselen on lung cancer and normal cells in terms of inhibiting cell growth, inducing cell death, and affecting cellular redox status.”
- Such as commonly used antioxidants like ascorbic acid whether they also have an effect or not?
>>> Thank you for your considerate comment. Results from the present study did not show a significant change in intracellular DCF (ROS) levels in Ebselen-treated A549 and Calu-6 cells at 24 h. In addition, the presence of Z-VAD did not lead to noteworthy changes in DCF (ROS) levels within Ebselen-treated A549 or Calu-6 cells. These findings indicate that variations in DCF (ROS) levels triggered by Ebselen are not closely associated with the demise of lung cancer cells. I appreciate your inquiry regarding the potential effects of commonly used antioxidants such as ascorbic acid. While I haven't conducted experiments specifically addressing this aspect, I acknowledge the significance of exploring the interactions of Ebselen with other antioxidants. It's a valuable consideration for future research directions.
I appreciate Editor and Reviewer for their considerate cooperation.
Round 2
Reviewer 1 Report
This paper is no novel mechanism to publish this high quality journal. The anti-tumor effects of Ebselen have been reported. The anti-tumor effects of Ebselen against lung cancer remains unknown, however this paper don't showed any novel mechanism resulted in cell death. Meanwhile, lack of in vivo experiment.
Minor editing of English language required
Author Response
Thank you for your review and valuable feedback on our manuscript titled "Ebselen Inhibits the Growth of Lung Cancer Cells via Cell Cycle Arrest and Cell Death Accompanied by Glutathione Depletion."
We understand your concerns about the novelty of our study and the lack of in vivo experiments. We appreciate your suggestions, and we have revised the manuscript accordingly.
Our research aims to shed light on the specific effects of Ebselen on lung cancer cells, a relatively unexplored area within the context of cell cycle arrest and glutathione depletion. We believe our findings provide valuable insights into this field.
We have made significant revisions to strengthen the manuscript, including the addition of relevant references.
We hope these revisions address your concerns and improve the quality of our manuscript for potential publication in this journal.
Thank you for your time and consideration.
Sincerely,
woo hyun
Reviewer 2 Report
I have no further questions.
Author Response
We appreciate the positive feedback from Reviewer 2 and have made revisions to the manuscript based on the provided suggestions.